# LncRNA NR120519 Blocks KRT17 to Promote Cell Proliferation and Migration in Hypopharyngeal Squamous Carcinoma

**DOI:** 10.3390/cancers15030603

**Published:** 2023-01-18

**Authors:** Zheng Zhou, Gehou Zhang, Tieqi Li, Jingang Ai, Wei Li, Shiyu Zeng, Maoyu Ye, Qian Liu, Jian Xiao, Yunqiu Li, Guolin Tan, Xiaowei Zhang

**Affiliations:** 1Department of Otolaryngology Head & Neck, Third Xiangya Hospital, Changsha 410013, China; 2Department of Otolaryngology Head and Neck Surgery, Hunan Provincial People’s Hospital, Changsha 410000, China

**Keywords:** hypopharyngeal squamous cell carcinoma, NR120519, KRT17, AKT/mTOR, EMT

## Abstract

**Simple Summary:**

Hypopharyngeal squamous cell carcinoma (HSCC) is a type of cancer with poor prognosis in head and neck tumors. More studies have shown that abnormal expression of lncRNA plays a crucial role in HSCC. Through RIP experiments, we confirmed that NR120519 interacts with KRT17, and the expression of both is closely related to the overall survival of HSCC. Subsequent experiments showed that both NR120519 and KRT17 could regulate the AKT/mTOR pathway and lead to EMT transformation, promoting the progression of HSCC. Therefore, NR120519/KRT17/AKT/mTOR axis has been identified as a new pathway, providing a feasible preliminary basis for future studies.

**Abstract:**

Background: Hypopharyngeal carcinoma is the worst type of head and neck squamous cell carcinoma. It is necessary to identify the key molecular targets related to the carcinogenesis and development of hypopharyngeal carcinoma. Methods: Differentially expressed lncRNAs in hypopharyngeal carcinoma were selected by microarray, and lncRNA-associated proteins were found by RIP assay. Colony formation, CCK-8, wound healing and Transwell assays were performed to detect the effects of lncRNA and its associated protein on cell proliferation and migration in vitro. Downstream pathways of lncRNA and its associated protein were detected by WB. Through a subcutaneous tumor model, the effects of lncRNA and its associated protein on cell proliferation were detected. The expressions of lncRNA and its associated protein in hypopharyngeal cancer tissues were detected by qRT-PCR and immunohistochemistry assays, respectively, and survival analyses were performed by Kaplan-Meier curve. Results: A total of 542 and 265 lncRNAs were upregulated and downregulated in microarrays, respectively. LncRNA NR120519 was upregulated and promoted cell proliferation and migration of hypopharyngeal carcinoma in vitro and cell proliferation in vivo. RIP and WB assays showed that KRT17 was associated with and blocked by NR120519.The silencing of KRT17 promoted cell proliferation and the migration of hypopharyngeal carcinoma in vitro and cell proliferation in vivo by activating the AKT/mTOR pathway and epithelial-mesenchymal transformation (EMT). Finally, the NR120519 high expression and KRT17 low expression groups showed shorter overall survival. Conclusion: NR120519 activated the AKT/mTOR pathway and EMT by blocking KRT17 to promote cell proliferation and the migration of hypopharyngeal carcinoma.

## 1. Introduction

Hypopharyngeal cancer refers to a special type of head and neck tumor characterized by higher malignancy than that of other head and neck tumors and similar pathogenesis and pathological features to those of esophageal cancer [1]. Most cases are squamous cell carcinoma and the majority of patients have a long history of smoking and alcohol consumption [2]. It has been found that hypopharyngeal cancer is closely related to gastroesophageal reflux, particularly the pepsin in the reflux component. Niu et al. conducted an immunohistochemical analysis and reported that the group composed of 70 patients with primary hypopharyngeal carcinoma had higher levels of pepsin expression than the control group [3]. This also suggests that the development of hypopharyngeal carcinoma may be associated with various types of irritation. Generally, hypopharyngeal carcinoma is detected late, with approximately 80% of cases diagnosed with stage III or IV of the disease. In a clinical study involving 595 patients with hypopharyngeal cancer, Hall et al. reported tumor recurrence in approximately 50% of cases. Most patients relapsed within 1 year after the end of treatment, and half of them showed distant metastases [4].

Tumor multicentricity and submucosal infiltration are also pathologic features of hypopharyngeal cancer; the latter is associated with positive surgical margins. Current treatment options for hypopharyngeal cancer include surgery, radiotherapy, chemotherapy and molecular targeted therapy. However, the overall therapeutic outcomes remain unsatisfactory. Patients with hypopharyngeal cancer are linked to a relatively low overall survival rate. Thus, it is important to reveal the molecular mechanisms underlying this disease and identify more sensitive and specific molecular markers involved in its progression. Such knowledge would potentially improve the rate of early diagnosis and prognosis of hypopharyngeal cancer.

A class of RNA transcripts with a length of more than 200 nucleotides is referred to as long non-coding RNA (lncRNA) [5]. lncRNAs typically do not encode proteins or peptides, and earlier research even dismissed them as being merely the waste products of gene expression. However, new research has linked the emergence of certain diseases to the unchecked expression of lncRNAs [6]. Disrupted lncRNAs play key roles in a variety of tumor pathogenic processes, including drug resistance, cell migration, invasion, and epithelial-mesenchymal transition (EMT) [7]. NR120519 is a lncRNA (length: 407 nucleotides) located on chromosome 7, which is upregulated in hypopharyngeal carcinoma. The NCBI Reference Sequence is available under NR_120519.1. Differentially expressed LncRNA see Appendix A. Moreover, It has been shown [8] that keratins are an important component of the cytoskeleton and are members of the intermediate filament superfamily. Keratins can be divided into two groups: 28 type I acidic proteins and 26 type II basic proteins. Keratin 17 (KRT17) belongs to the type I intermediate keratin family. KRT17 is a multifunctional protein that regulates many cellular processes, including cell proliferation and growth [9]. In normal healthy epithelia, KRT17 expression is restricted to the medullary compartments of hair and skin adnexa, while expression is abnormal in many types of cancer [10,11]. The objective of the present study was to investigate the potential relationship between NR120519 and KRT17 in hypopharyngeal carcinoma.

## 2. Materials and Methods

### 2.1. Tissue Samples

Sixty-five specimens of hypopharyngeal cancer and adjacent normal tissues surgically resected between 2015 and 2019 at the Third Xiangya Hospital of Central South University (Changsha, China) were collected. Part of each specimen was embedded in wax, while the remaining specimen was preserved in liquid nitrogen. Clinical staging was determined according to the classification established by the 7th International Conference on Otolaryngology staging. Written informed consent for all collected tissues was provided by the patients, and the study was approved by the Ethics Committee of the Third Xiangya Hospital of Central South University (Changsha, China).

### 2.2. Microarray

With the aid of Arraystar human lncRNA microarrays, the relative expression levels of lncRNA and mRNA were ascertained. Utilizing the TRIzolTM reagent, total RNA was isolated from three pairs of clinical samples (Invitrogen Life Technologies, Carlsbad, CA, USA). We carried out microarray hybridization in accordance with the manufacturer’s instructions. Agilent’s DNA microarray scanner was used to scan the hybridization arrays, and the company’s feature extraction software then collected the resulting data (V11.0.1.1; Agilent Technologies, Santa Clara, CA, USA). Using the Agilent GeneSpring GX software, quantification was carried out after data normalization (V12.1; Agilent Technologies). Low-intensity RNAs were removed and then lncRNAs and mRNAs that were identified as “present” or “borderline” in at least one of the six samples were chosen for additional examination. The criteria denoting significant expression of lncRNAs or mRNAs were *p*-values < 0.05 and z|log2 (Fc)| ≥ 1.5. Differentially expressed LncRNA see Appendix A.

### 2.3. Cell Culture and Transfection

Human pharyngeal carcinoma cells (FaDu and Detroit 562) and human nasopharyngeal epithelial cells (NP69) were obtained from the American Type Culture Collection (Manassas, VA, USA). Cells were cultured in a minimal essential medium (MEM) containing 10% fetal bovine serum (Every Green). The cell lines were identified by short tandem repeat sequence (STR). Genechem Co., Ltd. (Shanghai, China) was utilized to synthesize sequences for the knockdown of NR120519 (sh1NR120519 interference fragment: 5′-GGTGCTGTGCTCTGTGGTG-3′, sh2NR120519 interference fragment: 5′-GGGGATTTCCTCTGGCAAA-3′) and KRT17 (shKRT17 interfering fragment: 5′-CAGTCGCGTTTGCGACTGG-3′) in a lentivirus. FaDu cells and Detriot562 were transfected with the knockdown or control lentivirus (multiplicity of infection: 10) for 16 h. Screening with puromycin confirmed the production of stably transfected cell lines. The cells were cultured in a cell culture incubator at 37 °C and 5% CO_2_.

### 2.4. Quantitative Real-Time Polymerase Chain Reaction (qRT-PCR) Analysis

Cells were collected and total RNA was extracted using Tri Reator (Sigma–Aldrich, St. Louis, MO, USA). Total RNA was reverse transcribed using the Access Quick RT-PCR kit (Promega). Real-time fluorescence quantification of NR120519 and KRT17 was performed on a Roche qPCR instrument. Data were normalized to the levels of the housekeeping gene β-actin, and the relative mRNA expression was calculated using the 2^−ΔΔCt^ method. Primers were synthesized by RiboBio and were as follows: 5′-GGCUGCGGGUGCAUCUUAUTT (sense) and 5′-AUAAGAUGCACCCGCAGCCTT-3′ (antisense) for NR120519; 5′-GGT GGG TGG TGA GAT CAA TGT-3′ (sense) and 5′-CGC GGT TCA GTT CCT CTG TC-3′ (antisense) for KRT17; 5′-ACACTCCAGCTG GGGTGCTCTTCGGCAGCACA-3′ (sense) and 5′-AGGGGTCCGAGGTATTC-3′ (antisense) for U6; 5′ GTGGCCGAGGACTTTGATTG3′ (sense) and 5′ CCTGTAACAACGCATCTCATATT3′ (antisense) for β-actin.

The relative expression of NR120519 in 35 hypopharyngeal cancer tissues was determined by qRT-PCR. Kaplan–Meier curve survival analysis was performed using the β-actin-corrected ΔCt value and a cut-off value of 12 (values <12 and ≥12 denoted high and low expression, respectively).

### 2.5. Colony Formation Assay 

After trypsin digestion, HSCC cells (1 × 10^4^) were humidified in 6-well plates for 3 weeks. Colonies were fixed with 4% paraformaldehyde and stained with 0.1% crystal violet. The number of colonies was counted using a microscope.

### 2.6. Cell Counting Kit-8 (CCK-8) Assay

The stably transferred cells were seeded in 96-well culture plates at a density of 2 × 10^3^ cells/well. Following cell cultures for 24 h, 48 h, 72 h and 96 h, the effect of gene knockdown on cell proliferation was evaluated using the CCK-8 assay. Medium (100 µL) and CCK-8 (10 µL) solutions were added to each well, and the plate was placed in the incubator at 37 °C for 4 h. The absorbance was measured at 450 nm using an enzyme marker.

### 2.7. Wound Healing Assay

The treated cells (2 × 10^6^) were placed in a 6-well plate and incubated at 37 °C until they reached 100% confluence. A scratch was performed on the surface of the cell layer using a sterile pipette. After replacing the medium with 1% serum MEM, the cells were incubated for 24 h. Finally, images of cell migration before and 24 h after cell scratching were captured under a microscope (Leica, Germany) at ×10 magnification.

### 2.8. Transwell Assay

The migratory ability of cells was determined using a Transwell chamber (Tewksbury, MA, USA) (pore size: 8 μm). Briefly, 2 × 10^5^ cells in serum-free MEM culture medium (100 μL) were placed in the upper chamber; a culture medium containing 10% fetal bovine serum (600 μL) was added to the lower chamber. Cells were incubated at 37 °C and 5% CO_2_ for 24 h and stained with 0.5% crystal violet. The cells that remained in the upper chamber were removed. Images were captured under a microscope (Leica, Germany) at ×10 magnification.

### 2.9. Western Blotting

Total proteins were extracted from cell lysates according to the procedure for the extraction of proteins from tissue and cells. Protein concentrations were quantified using the bicinchoninic acid (BCA) method assay kit according to the instructions provided by the manufacturer. Equal amounts of protein from each sample were separated electrophoretically on a sodium dodecyl sulfate/polyacrylamide gel and transferred to polyvinylidene difluoride membranes (Roche). The polyvinylidene fluoride membranes were incubated with primary antibodies at 4 °C overnight. The primary antibodies used in this study were as follows: protein kinase B (AKT) rabbit primary antibody (1:1000; Beyotime, Shanghai, China); phosphorylated-AKT (p-AKT; Ser473) rabbit primary antibody (1:2000; Beyotime, Shanghai, China); mammalian target of rapamycin kinase (mTOR) mouse primary antibody (1:20,000; Proteintech, Wuhan, China); p-mTOR (Ser2448) mouse primary antibody (1:5000; Proteintech, Wuhan, China); E-cadherin rabbit primary antibody (1:1000; Cell Signaling Technology, Massachusetts, USA); N-cadherin rabbit primary antibody (1:1000; Affinity, Changzhou, China); vimentin (VIM) mouse primary antibody (1:20,000; Proteintech, Wuhan, China); and KRT17 rabbit primary antibody (1:1000; Proteintech, Wuhan, China). Glyceraldehyde-3-phosphate dehydrogenase (GAPDH) was used as the internal reference protein.

### 2.10. Immunohistochemistry

For the immunohistochemistry examination, a rabbit polyclonal antibody against KRT17 (1:1000; Proteintech) was utilized. Two qualified reviewers examined tissue samples without having access to patient information. Each slide was inspected five times, with 100 cells being visible at a 400× magnification in each view. Based on the percentage of positive cells and the degree of cell staining, the results were graded. The following criteria were used to define immunostaining intensity: Negative values were represented by 0, weak values by 1, moderate values by 2 and positive values by 3 (strong). The percentages of positive cells were as follows: 0 (5%), 1 (5–25%), 2 (26–50%), 3 (51–75%) and 4 (>75%). The two scores were multiplied to arrive at the final score for each case. A KRT17 value less than 6 indicated low expression, while a KRT17 value greater than or equal to 6 indicated high expression.

### 2.11. Fractionation of Nuclear/Cytoplasmic RNA

The Fadu cell precipitate was collected and a cell lysis buffer (50 mM Tris-HCl, pH 7.4, 0.14 M NaCl, 1.5 mM MgCl_2_, 0.5% NP-40, 1 U/uL RNase inhibitor, 1 mM DTT) (×20 the volume of the cell precipitate) was added. The mixture was centrifuged at 1500× *g* for 5 min.

Next, the supernatant was collected as the crude extract of the cell pulp. The supernatant was removed, an equal volume of cell lysis buffer was added and the mixture was centrifuged again at 1500× *g* for 5 min to obtain a precipitated fraction of cytoplasmic RNA. The residue remaining after cell pulp extraction was added to TRIZOL reagent, repeatedly puffed to lyse the nuclei, incubated for 5 min in order to completely dissociate the nucleic acid-protein complex, 0.2 mL of chloroform was added and the mixture was shaken for 15 s, incubated for 2–3 min and centrifuged for 15 min. The upper colorless aqueous phase was taken, half of isopropanol was added and mixed well, and the mixture was incubated for 10 min and centrifuged for 10 min, obtaining RNA at the bottom and side walls of the tubes for intra-nuclear RNA. qRT-PCR analysis was performed on the obtained cytoplasmic and cell membrane extracts to determine the level of NR120519.

### 2.12. RNA Immunoprecipitation (RIP) Assay

RIP assay was performed using the Magna RIPTM RNA-binding protein immunoprecipitation kit (microwells). Cells were lysed with RIP lysate, immunoprecipitated overnight with immunoglobulin G (IgG) antibody and argonaute RISC catalytic component 2 (Ago2) antibody encapsulated on magnetic beads, and washed with phosphate-buffered saline. A fraction of the cells was used as a negative control (termed INPUT). RNA in the co-precipitates was extracted with TRIzolTM and the purified RNA was analyzed by RT-qPCR. Significant proteins see Appendix A.

### 2.13. In Vivo Xenograft Experiments

Male BALB/c nude mice (age: 4–6 weeks; weight: 18–22 g) were purchased from Hunan Slack Jingda Co., Ltd. (Changsha, China). Subsequently, cell suspension (2 × 10^6^ cells in 100 μL) of FaDu, KRT17 and Detriot562 cells stably expressing sh1NR120519, sh2NR120519 or shKRT17, or negative controls (shNC) were injected into the axillary skin of mice (*n* = 4). Thirty days after injection, the mice were sacrificed, and the tumor volume was calculated as follows: volume (mm^3^) = 0.5 (width) × 2 (length). All animal experiments were approved by the Animal Ethics Committee of the Third Xiangya Hospital of Central South University (Changsha, China).

### 2.14. Statistical Analysis

The mean ± standard deviation of the data is displayed. A t-test was used to compare the two groups’ values for continuous variables. Multiple group comparisons were made using the analysis of variance (ANOVA). The software SPSS 25.0 (IBM Corp., Armonk, NY, USA) was used for all analyses. Differences with *p*-values < 0.05 were considered statistically significant.

## 3. Results

### 3.1. NR120519, Which Was Upregulated in Hypopharyngeal Carcinoma Tissues, Played a Tumor-Promoting Role

We selected three pairs of hypopharyngeal carcinoma and paraneoplastic tissues for microarray analysis to identify differentially expressed lncRNAs. The results showed that 542 and 265 lncRNAs were upregulated and downregulated, respectively (Figure 1A). Differentially expressed LncRNA see Appendix A. We verified some lncRNAs with z|log2 (Fc)| > 2 and *p* < 0.05 screened by microarray in 10 pairs of hypopharyngeal carcinoma and paracancer tissues. There were three lncRNAs with z|log2 (Fc)| > 2 in all verified specimens, but only NR120519 was functional after knockdown. We found that NR120519 expression exhibited an average 6.41-fold upregulation in hypopharyngeal carcinoma tissues (Figure 1B). The results of the qRT-PCR analysis showed that the knockdown efficiency of sh1NR120519 and sh2NR120519 in the hypopharyngeal carcinoma Fadu cell line was 53.74 ± 4.42% and 64.61 ± 0.60%, respectively (Figure 1C). Subsequently, In the Fadu cells, we performed colony formation and CCK-8 assays to assess the effects of NR120519 knockdown. The results of the colony formation assay showed that the number and relative area of colonies were significantly reduced after NR120519 knockdown (Figure 1D), while those of the CCK-8 assay demonstrated that cell proliferation was retarded (Figure 1E). The wound healing and Transwell assays showed that the migratory ability of cells was significantly reduced after NR120519 knockdown (Figure 1F,G). We performed subcutaneous tumorigenesis in nude mice. As shown in Figure 1H, the tumor size was significantly reduced in the knockdown NR120519 group compared with the control group (51.1 ± 16.6 mm^3^ and 184.1 ± 58.2 mm^3^, respectively). To clarify the cellular localization of NR120519, we collected Fadu cells for the separation of the nucleus and cytoplasm and performed qRT-PCR to verify the expression of NR120519. The results showed that 95.67 ± 0.78% of NR120519 was located in the nucleus (Figure 1I).

Similarly, the qRT-PCR analysis showed that the knockdown efficiency of sh1NR120519 and sh2NR120519 in the hypopharyngeal carcinoma Detroit 562 cell line was 50.36 ± 12.5% and 68.95 ± 4.19%, respectively (Figure 2A). We evaluated the effect of NR120519 knockdown by colony formation and CCK-8 assays. The results of the colony formation experiment showed that the number and relative area of colonies were significantly reduced after NR120519 gene knockdown. (Figure 2B). However, CCK-8 assay showed that cell proliferation was inhibited (Figure 2C). Wound-healing and Transwell analysis showed that NR120519 knockdown significantly reduced cell migration (Figure 2D,E).

### 3.2. NR120519 Inhibited KRT17 Expression via Activation of the AKT/mTOR and EMT Pathways

After pulling down NR120519 through RIP assay, we found that KRT17 bonded directly to NR120519 (Figure 3A,B). Significant proteins see Appendix A. Western blotting experiments revealed that KRT17 expression was upregulated by 1.87 ± 0.04-fold in the knockdown group compared to the control group. KRT17 has been reported to be closely linked to the AKT/mTOR and EMT pathways. Therefore, in the Fadu cell lines, we examined the expression of the above pathways after knockdown of NR120519. The results showed that AKT and mTOR phosphorylation levels were significantly decreased, E-cadherin expression was upregulated, and N-cadherin and VIM expression was downregulated in the knockdown group versus the control group (Figure 3C).

Similarly, in the Detroit 562 cell line, we detected the expression of the above pathways after knockdown of NR120519. The results showed that AKT and mTOR phosphorylation levels were significantly decreased, E-cadherin expression was up-regulated, and N-cadherin and VIM expression were down-regulated in the knockdown group compared with the control group (Figure 4A). In addition, we detected increased expressions of KRT17 in the HSCC cell lines, FaDu and Detroit 562, and low expressions in the normal human nasopharyngeal epithelial cell line, NP69 (Figure 4B).

### 3.3. KRT17 Exerted Anticancer Effects on Hypopharyngeal Cancer and Inhibited the AKT/mTOR and EMT Pathways

We used lentivirus to knock down KRT17 in the hypopharyngeal carcinoma Fadu cell line. The results of western blotting experiments showed that the expression of KRT17 in the knockdown group was only 30.1 ± 1.0% of that observed in the control group, and we verified the AKT/mTOR and EMT pathways. The results showed that the phosphorylation levels of AKT and mTOR were significantly increased, the expression of E-cadherin was downregulated, and the expression of N-cadherin and VIM was upregulated in the knockdown group versus the control group (Figure 5A). We performed colony formation and CCK-8 assays using knockdown and control cells to determine their growth ability. The results of the colony formation assay showed that the number of colonies was not significantly different after knockdown of KRT17; however, the relative area was significantly increased (Figure 5B). In addition, the results of the CCK-8 assay showed that cell proliferation was increased after knockdown of KRT17 (Figure 5C). Next, we performed scoring and Transwell experiments, which showed that knockdown of KRT17 significantly increased the migratory ability of cells (Figure 5D,E). We also performed subcutaneous tumorigenesis experiments in nude mice. The results showed that the tumor size was significantly increased in the KRT17 knockdown group compared to the control group (186.3 ± 43.2 mm^3^ and 67.8 ± 27.2 mm^3^, respectively) (Figure 5F).

We also noticed that tumor proliferation was significant and the ability to invade and metastasize was substantially increased in the Detroit 562 cell line with KRT17 knockout compared to the control group (Figure 6). On the contrary, the proliferation and metastasis ability of tumor cells were significantly decreased in Fadu and Detroit 562 cell lines overexpressed with KRT17 (Figure 7).

### 3.4. Knockdown of NR120519 Reversed the Effect of KRT17 Knockdown

We subsequently knocked down NR120519 in KRT17-knockdown Fadu cells. The results of western blotting experiments showed that, compared to the control group, the knockdown of NR120519 resulted in an approximately 2-fold increase in KRT17 expression, a significant decrease in AKT and mTOR phosphorylation levels, an upregulation of E-cadherin expression, and a downregulation of N-cadherin and VIM expression (Figure 8A). The results of colony formation experiments showed that the number and relative area of colonies were significantly reduced after knocking down NR120519 (Figure 8B). CCK-8 assays demonstrated that knockdown of NR120519 retarded the proliferation of cells (Figure 8C). Next, wound healing and Transwell experiments showed that knockdown of NR120519 significantly decreased the migratory ability of cells (Figure 8D,E). We also performed subcutaneous tumorigenesis experiments in nude mice, and the results are shown in Figure 8F. Compared to the control group, the NR120519 knockdown group exhibited a significantly reduced tumor size (318.6 ± 93.9 mm^3^ and 76.6 ± 22.5 mm^3^, respectively).

We also knocked down NR120519 in addition to KRT17 in Detroit 562 cells. Western blotting showed that compared to the control group, the expression of KRT17 was increased 1.48 ± 0.12 times after NR120519 gene knockout, the phosphorylation of AKT and mTOR was significantly decreased, and the expression of E-cadherin was up-regulated. The expressions of N-cadherin and VIM were down-regulated (Figure 9A). The results of colony formation experiments showed that the number and relative area of colonies were significantly reduced after knocking down NR120519 (Figure 9B). CCK-8 assays showed that knockdown of NR120519 delayed cell proliferation (Figure 9C). Next, wound-healing and Transwell assays showed that knockdown of NR120519 significantly reduced cell migration (Figure 9D,E).

### 3.5. High Expression of NR120519 and KRT17 Denoted Worse and Better Prognosis, Respectively

We collected 35 fresh tissues of hypopharyngeal cancer and classified them into high and low-expression groups according to NR120519 levels (detected by qRT-PCR). Kaplan–Meier survival analysis showed that the high-expression group was associated with shorter overall survival than the low-expression group (Figure 10A). We used the Gene Expression Profiling Interactive Analysis (GEPIA) database analysis to determine that KRT17 was highly expressed in head and neck malignancies compared with paraneoplastic tissues (Figure 10B). We collected 65 tissues of hypopharyngeal cancer embedded in wax blocks for immunohistochemical experiments; among those, 10 cases had paraneoplastic controls. We found that KRT17 was highly expressed in cancer tissues relative to paraneoplastic tissues (Figure 10C). For survival analysis, we classified the tissues into high and low-expression groups according to the expression of KRT17. The results showed that the high-expression group had longer overall survival than the low-expression group (Figure 10D).

## 4. Discussion

Advances have been achieved in surgical modalities, intensity-modulated radiotherapy and the concurrent use of radiotherapy and targeted drug therapy against hypopharyngeal cancer. Nevertheless, the outcome of treatment remains unsatisfactory, and the 5-year survival rate of patients has not shown significant improvement. Recently, a study showed that the 5-year survival rate of patients with advanced hypopharyngeal cancer who received radiation therapy was only 9% higher than that of patients who underwent surgery alone [12]. In the literature, it has been noted that the combination of surgery with radiotherapy has improved the prognosis and survival of patients with advanced hypopharyngeal cancer [13]. However, these effects were limited. Moreover, sequelae of exposure to radiation, the high toxicity of chemotherapy drugs and the loss of function caused by surgery reduce patients’ quality of life and impose a heavy psychological burden on them. Importantly, this disease is associated with short-term recurrence and metastasis. These facts highlight the shortcomings of current treatments and the importance of prevention and more precise treatment. Therefore, the identification of key prognostic predictors and targets for the treatment of hypopharyngeal cancer is crucial.

Numerous studies have suggested that lncRNAs play significant roles in various tumors, showing great temporal, spatial and tissue specificity [14,15,16,17,18,19,20]. LncRNAs play oncogenic or suppressive roles in different tumors, even at different stages in the same tumors. For example, a conserved long non-coding RNA (lncRNA) that is widely and abundantly expressed in human tissues and cell lines is non-coding RNA activated by DNA damage (NORAD). In order to segregate PUMILIO (PUMILIO RNA binding family member) proteins, control mitosis and preserve genomic stability, Lee et al. demonstrated that NORAD functions as a molecular ruse [21]. In contrast, metastasis-associated lung adenocarcinoma transcript 1 (MALAT1) interacts with the RNA-binding protein heterogeneous ribonucleoprotein C (C1/C2) (HNRNPC), translocates to the cytoplasm in the G2/M phase and promotes cell cycle progression [22]. Taurine upregulated 1 (TUG1) expression is upregulated in most tumor tissues but downregulated in laryngeal carcinoma and LAD tissues, suggesting that it may have tissue-specific expression patterns and functions in different types of tumors in humans [23]. AFAP1 antisense RNA 1 (AFAP1-AS1) was upregulated in non-small cell lung cancer tissues and cell lines [24]. In non-small cell lung cancer cells cultured in vitro, the inhibition of AFAP-AS1 inhibits growth and migration, whereas it promotes apoptosis. HOX transcript antisense RNA (HOTAIR) is one of the first lncRNAs linked to carcinogenesis. The expression levels of HOTAIR are elevated in a variety of tumors, including breast cancer, lung cancer, GC, hepatocellular carcinoma and glioma [25,26,27,28,29,30].

In hypopharyngeal carcinoma, it is also important to identify such functional lncRNAs and their corresponding roles. Gene microarray analysis showed that the expression of NR120519 was upregulated in cancer cells. In vivo and in vitro experiments involving knockdown of NR120519 in Fadu cells revealed its pro-cancer effect. We performed RIP assays and found that NR120519 binds directly to KRT17 proteins. Hence, we hypothesized that NR120519 may act by regulating KRT17 in hypopharyngeal carcinoma. Further study revealed that NR120519 is mainly localized in the nucleus, while KRT17 was capable of nuclear localization. These findings suggested that NR120519 may block the action of KRT17 by inducing the nuclear localization of KRT17 proteins. We showed an increase in KRT17 mRNA expression after knocking down NR120519, which also suggests that NR120519 regulates the transcription of KRT17.

Further supporting our theory, we discovered through additional in vivo and in vitro investigations that KRT17 has an antitumor effect in hypopharyngeal cancer. Keratin is a member of the intermediate filament superfamily that makes up the cytoskeleton and is encoded by 54 evolutionarily conserved genes. Based on gene substructure and nucleotide sequence homology, keratins can be classified into two groups: 28 type I acidic proteins and 26 type II basic proteins [31]. KRT17 is a type I keratin that is broadly dispersed in epithelial cells, according to earlier research [32]. Normal skin does not display it in the epidermis, but stress-related factors, such as scratching the skin, might cause it. KRT17 is considered to work in the cytoplasm. Recent research, however, suggested that KRT17 can enter and exit the nucleus since nuclear localization signals and nuclear export signals are present [33]. This novel idea of an intermediate filament located in the intranucleus raises the prospect that keratin controls further cellular functions. This shows that NR120519 might induce KRT17 into the nucleus and inhibit its activity in order to have a pro-oncogenic effect. The immunohistochemical study further demonstrated the tumor suppressive function of KRT17, demonstrating that high KRT17 expression was positively connected with the prognosis of patients with hypopharyngeal carcinoma and that KRT17 was substantially expressed in cancer tissues.

Many studies have suggested that KRT17 exerts a pro-carcinogenic effect in other types of cancer (e.g., lung cancer [34], Ewing’s sarcoma [35], pancreatic cancer [36], esophageal cancer [9] and oral squamous carcinoma [8]). However, our experimental results showed the opposite function in hypopharyngeal cancer, indicating the tissue specificity of KRT17. Interestingly, Quinn et al. used single-cell profiling with CRISPR-associated protein 9 (Cas9) to follow the metastatic path of mouse lung cancer cells. They discovered that KRT17 greatly reduced the capacity of lung cancer cells to spread [37]. Li et al. also demonstrated that the downregulation of KRT17 led to the loss of E-cadherin and altered EMT and the metastatic behavior of GC cells. It was suggested that loss of KRT17 led to the reorganization of the cytoskeleton, further activating Yes-associated protein (YAP) signaling and increasing interleukin 6 (IL6) expression, thus enhancing the metastatic ability of GC cells [38]. According to GEPIA2, high expression of KRT17 in most tumors was associated with poor prognosis, while high expression of KRT17 in BRCA was associated with good prognoses. These results suggest that KRT17 may play a role as a cancer suppressor gene in BRCA; as to the exact mechanism, it is not clear [39]. Similarly, in hypopharyngeal carcinoma, we found that KRT17 acts as a cancer suppressor gene; this effect may be related to the tissue specificity of KRT17. It is well established that hypopharyngeal carcinoma is a relatively special type of head and neck tumor, and its pathogenesis differs from those of other head and neck squamous carcinomas. Since KRT17 expression is induced during skin wound healing, we hypothesize that its expression may be related to various types of stimuli received by hypopharyngeal mucosa and tissues. Following stimulation, the hypopharyngeal epithelium is damaged, the balance of intermediate filament proteins is disrupted, keratin appears to be reorganized and its expression is induced, and KRT17 expression is elevated. Following the downregulation of KRT17 expression, the cytoskeletal stability, internal structure of cells and intercellular adhesion are impaired, leading to enhanced cell invasion and metastasis. In the present study, the reduction of E-cadherin expression in Fadu and Detroit 562 cells after knockdown of KRT17 supported this claim.

Tumors of epithelial origin are characterized by carcinogenesis typically involving EMT, which is a key mechanism leading to the acquisition of aggressive and malignant epithelial cancer cells [40]. Recent studies have revealed that KRT17 can inhibit tumor cell proliferation, migration and invasion in malignant tumors by regulating the Akt/mTOR pathway, glucose uptake [41], Wnt signaling pathway, epithelial-mesenchymal transition (EMT) [42] and mTOR/S6K1 signaling pathway [43]. In oral cancer, high expression of KRT17 induces EMT through the activation of the AKT pathway [44,45]. We found that the activation of the AKT pathway is closely related to the development of hypopharyngeal carcinoma. However, whether the low expression of KRT17 is related to the activation of the mTOR/AKT pathway has not been investigated. We found that knockdown of KRT17 expression upregulated the expression of p-mTOR, p-AKT, N-cadherin and VIM, whereas it downregulated that of E-cadherin. Moreover, knockdown of NR120519 upregulated the expression of KRT17 and E-cadherin and downregulated that of p-mTOR, p-AKT, N-cadherin and VIM. The binding of NR120519 to KRT17 may have blocked KRT17. Knockdown of both NR120519 and KRT17 downregulated the expression of p-mTOR, p-AKT, N-cadherin and VIM, whereas it upregulated that of E-cadherin. These findings reinforce the notion that NR120519 induced the transformation of hypopharyngeal carcinoma EMT by blocking the function of KRT17 through the activation of the mTOR/AKT pathway. Through immunohistochemical experiments, we examined the pathological sections of 65 patients with hypopharyngeal carcinoma. We found that the expression of KRT17 and NR120519 was positively and negatively correlated with the prognosis of patients with hypopharyngeal carcinoma, respectively.

The present study has several limitations. Firstly, the clinical sample size was small, and a sufficient number of BALB/c nude mice per group was required to improve the confidence of the results. Secondly, although the biological functions of KRT17 in vitro and in vivo were confirmed, the mechanisms involved in these effects remain to be investigated in depth. Therefore, further studies are warranted to validate the mechanism of the action of KRT17 in Fadu and Detroit 562 cells. This mechanism may involve specific proteins associated with cell adhesion, cell cycle, apoptosis, drug resistance and other complex processes.

In summary, this study showed that NR120519 expression was significantly upregulated in hypopharyngeal carcinoma tissues, consistent with the expression of KRT17. High expression of NR120519 and low expression of KRT17 activated the mTOR/AKT signaling pathway and induced stronger proliferation and migration of Fadu cells in vitro and in vivo. Most importantly, both high expression of NR120519 and low expression of KRT17 were closely associated with enhanced malignancy and poor prognosis of hypopharyngeal carcinoma. We further determined that NR120519 exerted a pro-carcinogenic effect by blocking the function of KRT17, while NR120519 and KRT17 were highly expressed in Fadu-DDP7-resistant cells constructed in our laboratory. This evidence may assist in guiding the individualized treatment of hypopharyngeal cancer and discovering new mechanisms of drug resistance.

## 5. Conclusions

LncRNA NR120519 was upregulated in hypopharyngeal cancer. NR120519 activated the AKT/mTOR pathway and EMT by blocking KRT17 and promoted cell proliferation and migration of hypopharyngeal carcinoma in vitro and cell proliferation in vivo. High expression of NR120519 and low expression of KRT17 both positively correlated to poor prognosis.

## Figures and Tables

**Figure 1 cancers-15-00603-f001:**
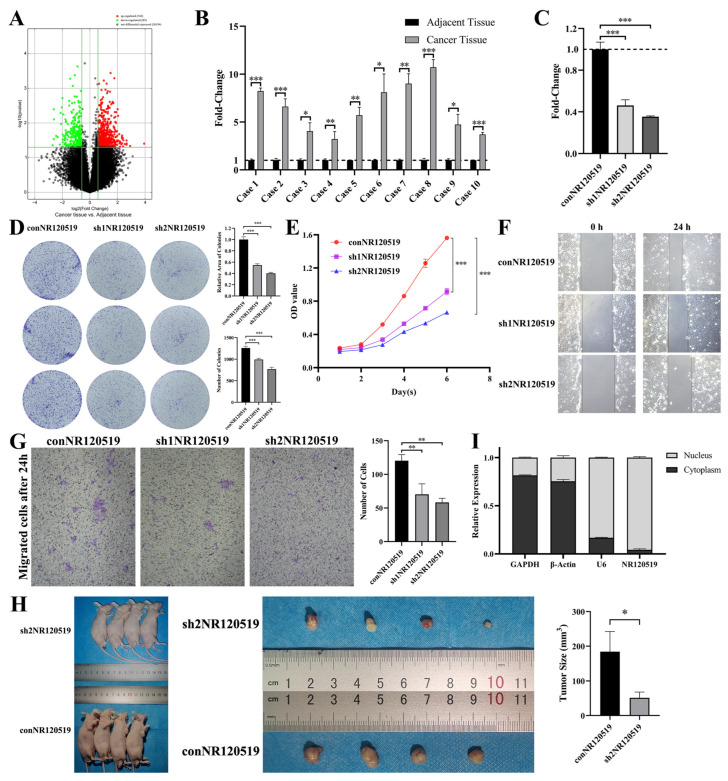
NR120519 was upregulated in hypopharyngeal carcinoma tissues and played a pro-carcinogenic role. (**A**) Volcano plots of differentially expressed lncRNAs in hypopharyngeal carcinoma and paraneoplastic tissues. (**B**) Relative expression of NR120519 in 10 pairs of hypopharyngeal carcinoma and paraneoplastic tissues. (**C**) Knockdown efficiency of two NR120519 targets (sh1NR120519 and sh2NR120519). (**D**) Colony formation assay involving NR120519 control and two knockdown groups. The cells were cocultured in 6-well plates. (**E**) CCK-8 experiments involving NR120519 control and two knockdown groups. (**F**) Wound healing experiments involving NR120519 control and two knockdown groups under ×10 magnification. (**G**) Transwell experiments involving NR120519 control group and two knockdown groups under ×10 magnification. (**H**) Tumorigenesis experiment in nude mice involving NR120519 control and knockdown (sh2NR120519) groups. (**I**) Relative expression of NR120519 in the cytoplasm and nucleus. * *p* < 0.05; ** *p* < 0.01; *** *p* < 0.001.

**Figure 2 cancers-15-00603-f002:**
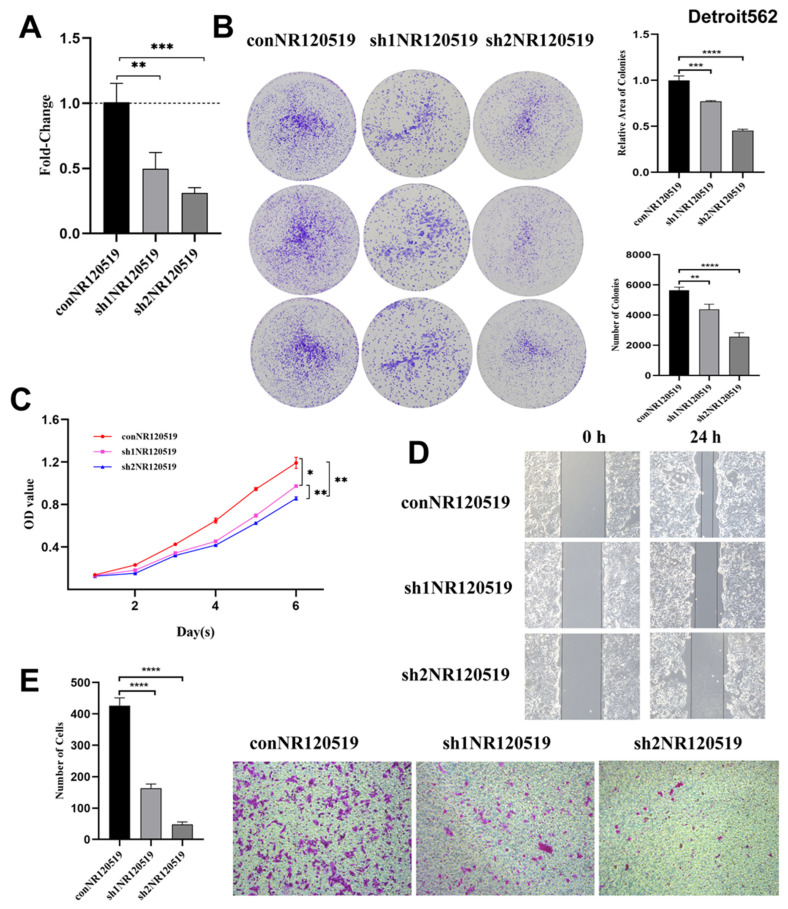
NR120519 exerted oncogenic effects and inhibited the AKT/mTOR and EMT pathways in hypopharyngeal carcinoma. (**A**) qRT-PCR analysis showed that the knockdown efficiency of sh1NR120519 and sh2NR120519 in hypopharyngeal carcinoma Detroit562 cell line was 50.36 ± 12.5% and 68.95 ± 4.19%, respectively. (**B**) Colony formation assay involving NR120519 control and knockdown (sh1NR120519 and sh2NR120519) groups. The cells were cocultured in 6-well plates. (**C**) CCK-8 assay involving NR120519 control and knockdown (sh1NR120519 and sh2NR120519) groups. (**D**) Wound healing experiments involving NR120519 control and knockdown (sh1NR120519 and sh2NR120519) groups under ×10 magnification. (**E**) Transwell assay involving NR120519 control and knockdown (sh1NR120519 and sh2NR120519) groups under ×10 magnification. * *p* < 0.05; ** *p* < 0.01; *** *p* < 0.001; **** *p* < 0.0001.

**Figure 3 cancers-15-00603-f003:**
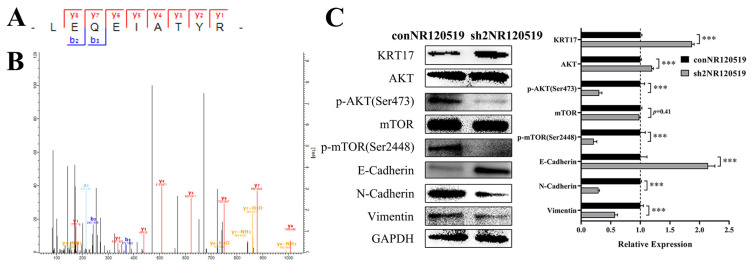
NR120519 repressed KRT17 expression and activated the AKT/mTOR and EMT pathways. (**A**) The ions determining the position of KRT17 in the RNA pulldown assay of NR120519. (**B**) Mass spectra of KRT17 protein after pulldown in the RNA pulldown assay of NR120519. (**C**) Western blotting experiments involving NR120519 control and knockdown (sh2NR120519) groups to detect the expression of KRT17, AKT/mTOR, and EMT pathway-related proteins. *** *p* < 0.001.

**Figure 4 cancers-15-00603-f004:**
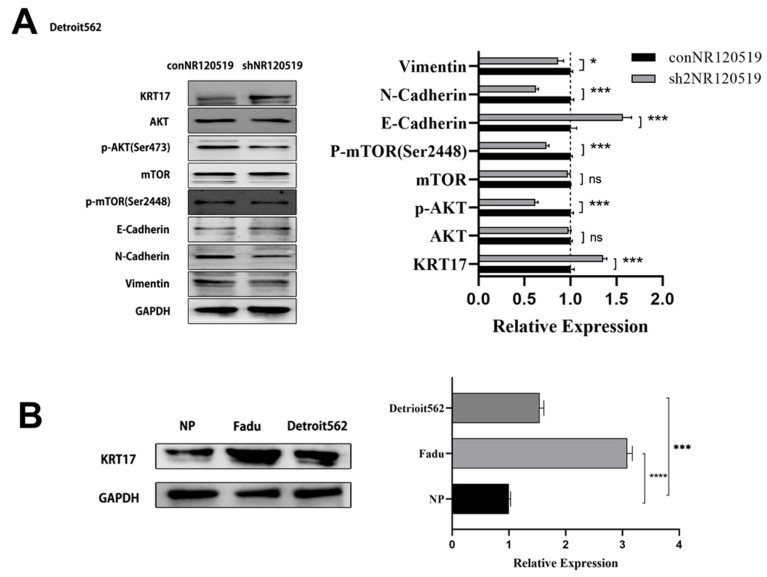
Knockdown of NR120519 in Detroit562 cell line inhibits AKT/mTOR pathway and increases KRT17 expression. (**A**) Western blotting experiments involving knockdown of NR120519 (sh2NR120519) and control (conNR120519 groups to detect the expression of KRT17, AKT/mTOR, and EMT pathway-related proteins. (**B**) Western blotting assay to detect KRT17 expression in NP, Fadu and Detroit562 cell lines. * *p* < 0.05; *** *p* < 0.001; **** *p* < 0.0001; ns no significance. The uncropped bolts are shown in Appendix A.

**Figure 5 cancers-15-00603-f005:**
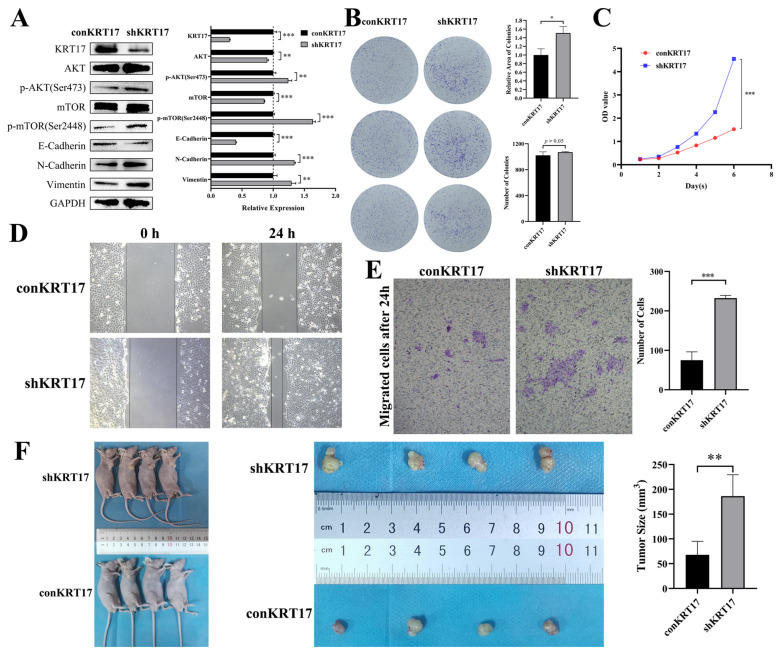
KRT17 exerted Anticancer effects on hypopharyngeal cancer and inhibited the AKT/mTOR and EMT pathways. (**A**) Phosphorylation levels of AKT and mTOR were significantly increased, E-cadherin was downregulated, N-cadherin and VIM was upregulated in the knockdown group versus the control group. (**B**) Colony formation assay showed that the number of colonies was not significantly different after knockdown of KRT17; however, the relative area was significantly increased. (**C**) CCK-8 assay showed that the cell proliferation was increased after knockdown of KRT17. (**D**,**E**) Scoring and Transwell experiments showed that knockdown of KRT17 significantly increased the migratory ability of tumor cells. The figures were captured under 10 × 10 microscope. (**F**) Tumorigenesis experiments involving KRT17 control and knockdown (shKRT17) groups in nude mice. * *p* < 0.05; ** *p* < 0.01; *** *p* < 0.001.

**Figure 6 cancers-15-00603-f006:**
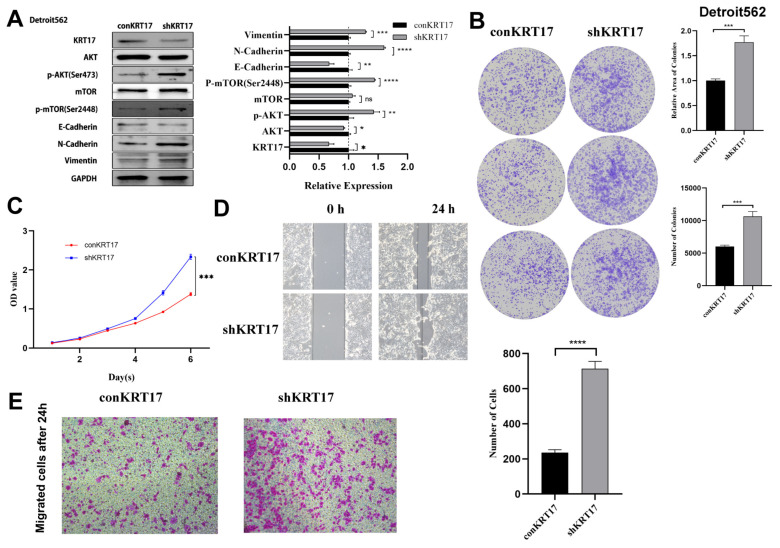
KRT17 knock down in Detroit562 cell line. (**A**) Phosphorylation levels of AKT and mTOR were significantly increased, E-cadherin was downregulated, N-cadherin and VIM was upregulated in the knockdown group versus the control group. (**B**) Colony formation assay showed that the number of colonies was not significantly different after knockdown of KRT17; however, the relative area was significantly increased. (**C**) CCK-8 assay showed that the cell proliferation was increased after knockdown of KRT17. (**D**,**E**) Transwell experiments showed that knockdown of KRT17 significantly increased the migratory ability of tumor cells. The figures were captured under 10 × 10 microscope. * *p* < 0.05; ** *p* < 0.01; *** *p* < 0.001; **** *p* < 0.0001; ns no significance. The uncropped bolts are shown in Appendix A.

**Figure 7 cancers-15-00603-f007:**
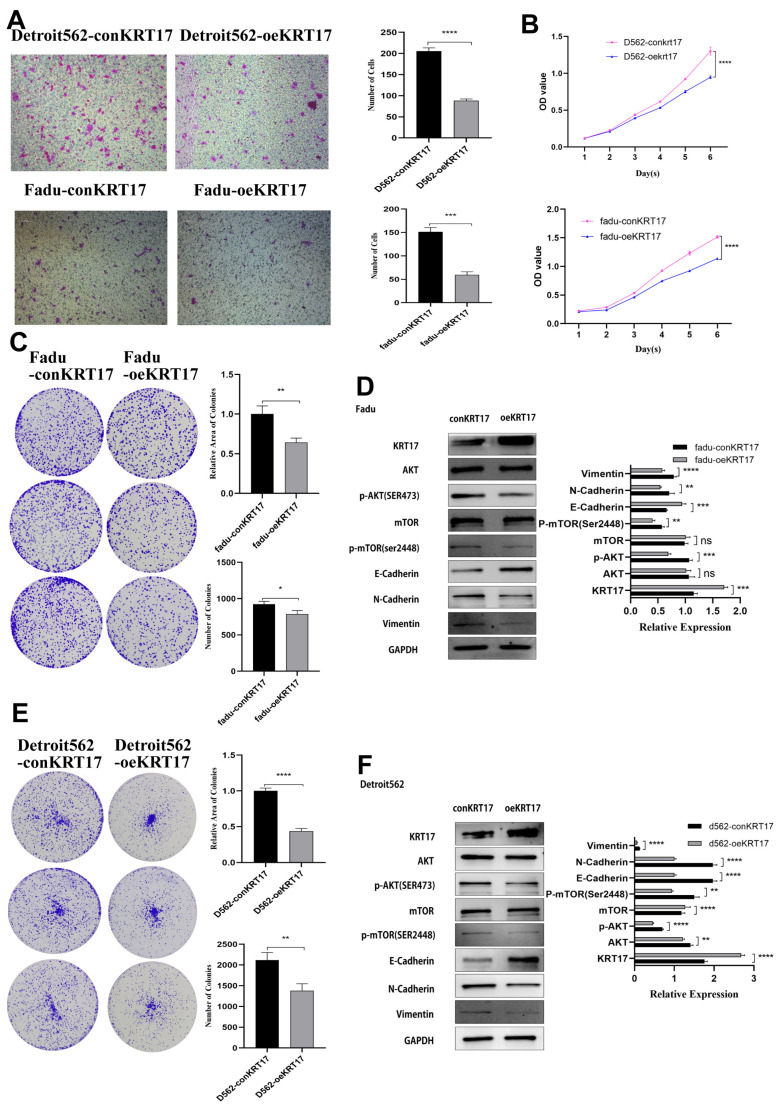
The proliferation and metastasis ability of tumor cells were significantly decreased in Fadu and Detroit562 cell lines overexpressed with KRT17. (**A**,**B**) Fadu and Detroit562 cell proliferation was inhibited after overexpressed with KRT17 showed by transwell and CCK-8 assay. (**C**,**D**) Colony formation assay and western blotting of Fadu cells after overexpressed with KRT17. (**E**,**F**) Colony formation assay and western blotting of Detroit562 cells after overexpressed with KRT17. The figures were captured under 10 × 10 microscope. * *p* < 0.05; ** *p* < 0.01; *** *p* < 0.001; **** *p* < 0.0001; ns no significance. The uncropped bolts are shown in Appendix A.

**Figure 8 cancers-15-00603-f008:**
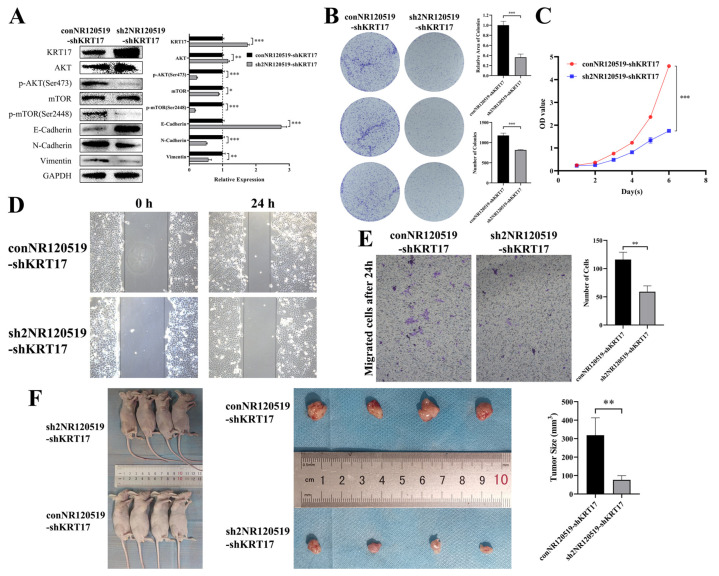
Knockdown of NR120519 reversed the effect of KRT17 silencing in Fadu cells. (**A**) western blotting showed that, the knockdown of NR120519 resulted in about 2 fold increase in KRT17 expression, significant decrease in AKT and mTOR phosphorylation levels, upregulation of E-cadherin, downregulation of N-cadherin and VIM expression. (**B**) Colony formation assays showed that Fadu colonies were significantly reduced after knocking down NR120519. (**C**) CCK-8 assays demonstrated that knockdown of NR120519 retarded the proliferation of cells. (**D**,**E**) wound healing and Transwell experiments showed that knockdown of NR120519 significantly decreased the migratory ability of cells. (**F**) NR120519 knockdown exhibited a significantly reduced tumor size. * *p* < 0.05; ** *p* < 0.01; *** *p* < 0.001. The uncropped bolts are shown in Appendix A.

**Figure 9 cancers-15-00603-f009:**
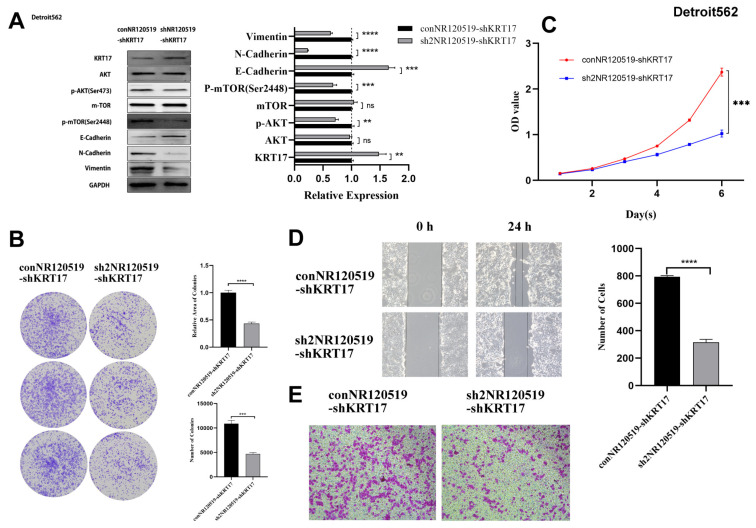
NR120519 knock down in Detroit562 cell line. (**A**) Western blotting showed that compared with the control group, the expression of KRT17 was increased 1.48 ± 0.12 times after NR120519 gene knockout, the phosphorylation of AKT and mTOR was significantly decreased, and the expression of E-cadherin was up-regulated. The expressions of N-cadherin and VIM were down-regulated. (**B**) Colony formation experiment showed that the number and relative area of colonies were significantly reduced after knocking down NR120519. (**C**) CCK-8 assay showed that knockdown of NR120519 delayed cell proliferation. (**D**,**E**) Wound-healing and Transwell assays showed that knockdown of NR120519 significantly reduced cell migration. The figures were captured under 10 × 10 microscope. ** *p* < 0.01; *** *p* < 0.001; **** *p* < 0.0001; ns no significance. The uncropped bolts are shown in Appendix A.

**Figure 10 cancers-15-00603-f010:**
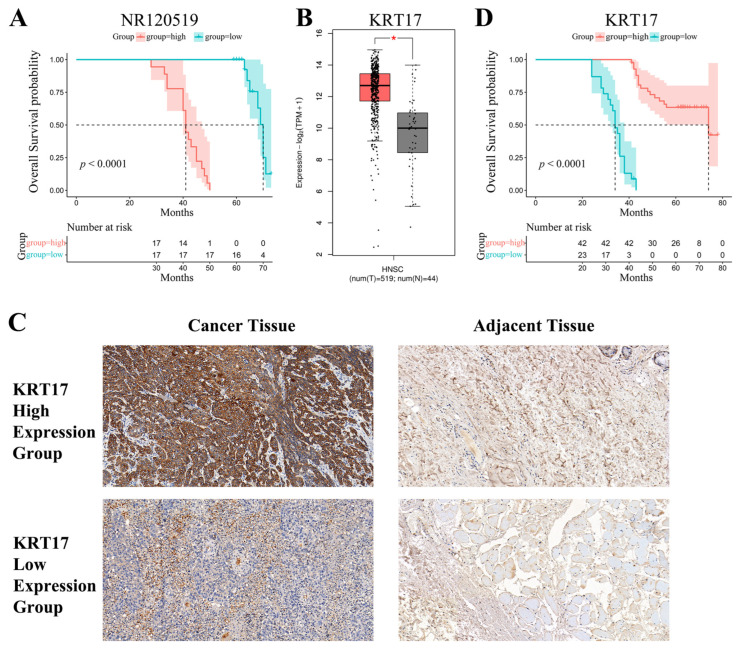
High expression of NR120519 and KRT17 denoted worse and better prognosis, respectively. (**A**) Kaplan–Meier survival analysis using relative Ct values of NR120519 from 35 fresh tissues of hypopharyngeal cancer. (**B**) The relative expression of KRT17 in head and neck malignancies versus paraneoplastic tissues was analyzed using the GEPIA database. (**C**) Immunohistochemical assay showing the expression of KRT17 in hypopharyngeal carcinoma tissues and paraneoplasms classified into high and low expression groups. Representative images (400×) are shown. (**D**) Immunohistochemical experiments were performed in 65 hypopharyngeal cancer tissues embedded in wax blocks, which were classified into KRT17 high and low expression groups. Kaplan–Meier survival analysis was performed. * *p* < 0.05.

## Data Availability

The data used to support the findings of this study are available from the corresponding author upon request.

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
