# Peer review of "LncRNA NR120519 Blocks KRT17 to Promote Cell Proliferation and Migration in Hypopharyngeal Squamous Carcinoma"

_cancers, 2023, doi:10.3390/cancers15030603_

Round 1

Reviewer 1 Report

The manuscript by Zhou et al describes a study about LncRNA NR120519 blocks KRT17, resulting in promoting hypopharyngeal squamous cancer cell proliferation and migration. I believe LncRNA studies have to be encouraged more for the future clinical trial. So it is valuable to find new LncRNAs to target genes.

However, there are a few concerns I would like recommend to add or revise. Some statements are also   confusing.

The Major concerns:

1. KRT17 has been studied as an oncogenic protein in various cancer cells through many other studies. Also, the authors discussed that KRT17 acts like oncogenic protein in head and neck cancer in Lines 75-78 and 436-447. However, the whole data here shows that KRT17 acts as a tumor suppressor protein. Many proteins could be controversial in different cancer cells. However, the statements have to be clear what the authors really want to state. Moreover, more references have to be added to support their conclusion. More statements about the KRT17 also have to be mentioned in Introduction and discussed in Discussion parts.

2. The figure legends are not shown in the PDF file shown to me.

3. To claim that KRT17 also acts as a tumor suppressor, KRT17 overexpression in the same cancer cell line should be performed.

The minor concerns:

1. In the material and method 2.4, more detail is required about the method of Nucleus and cytoplasm separation for RNA.

2. In the material and method 2.12, more detail is required.

3. There is no information about Fadu cell line including its original source in material methods.

4. Why paraneoplastic tissues used as a pair of hypopharyngeal carcinoma in Line 221? Because this tissue is also considered as cancerous tumors. More detailed information or explanation about this part should be added. For example, why this tissues were used, if these samples are from the same patient, what paraneoplastic tissue here exactly means and its location.

5. The form for the reference insert in the manuscript is not correct.

Author Response

Dear Reviewers:

Thank you very much for your kindly comments on our manuscript. Based on your and reviewer’s suggestions, we carefully revised the manuscript.

We are now sending the revised article for your re-consideration to publish in Cancers. Please see our point-to-point responses to all your comments below, and the corresponding revisions in the body of manuscript, marked in blue and red, respectively. We look forward to hearing from you soon for a favorable decision.

Thank you again for your time and consideration.

Sincerely,

Xiaowei Zhang

Reviewer 1#

Comments and Suggestions for Authors

The manuscript by Zhou et al describes a study about LncRNA NR120519 blocks KRT17, resulting in promoting hypopharyngeal squamous cancer cell proliferation and migration. I believe LncRNA studies have to be encouraged more for the future clinical trial. So it is valuable to find new LncRNAs to target genes.

However, there are a few concerns I would like recommend to add or revise. Some statements are also   confusing.

The Major concerns:

  1. KRT17 has been studied as an oncogenic protein in various cancer cells through many other studies. Also, the authors discussed that KRT17 acts like oncogenic protein in head and neck cancer in Lines 75-78 and 436-447. However, the whole data here shows that KRT17 acts as a tumor suppressor protein. Many proteins could be controversial in different cancer cells. However, the statements have to be clear what the authors really want to state. Moreover, more references have to be added to support their conclusion. More statements about the KRT17 also have to be mentioned in Introduction and discussed in Discussion parts.

Response: Thank you for your comments. As for the oncogenic protein effect of KRT17 in head and neck tumors, I would like to explain that KRT17 only plays a oncogenic protein role in oral cancer and esophageal cancer, and a tumor suppressor protein effect in hypopharyngeal cancer in our study. However, there have been no literature reports on other head and neck tumors, such as nasopharyngeal cancer and laryngeal cancer. This reflects the specificity of KRT17 in various cancer tissues.Besides, the application of KRT17 in head and neck cancer is also discussed in the Introduction by referring to last literature. Please check the details in the manuscript.

  1. The figure legends are not shown in the PDF file shown to me.

Response: Thank you for your comments. Please check the details in the last manuscript.

  1. To claim that KRT17 also acts as a tumor suppressor, KRT17 overexpression in the same cancer cell line should be performed.

Response: Thank you for your comments. Experiments on KRT17 overexpression are already under way, and due to the time required for these experiments, it may be a while longer before they are presented.

The minor concerns:

  1. In the material and method 2.4, more detail is required about the method of Nucleus and cytoplasm separation for RNA.

Response: Thank you for your comments. Please check the details in the manuscript.

  1. In the material and method 2.12, more detail is required.

Response: Thank you for your comments. Please check the details in the manuscript.

  1. There is no information about Fadu cell line including its original source in material methods.

Response: Thank you for your comments. Please check the details in the manuscript.

  1. Why paraneoplastic tissues used as a pair of hypopharyngeal carcinoma in Line 221? Because this tissue is also considered as cancerous tumors. More detailed information or explanation about this part should be added. For example, why this tissues were used, if these samples are from the same patient, what paraneoplastic tissue here exactly means and its location.

Response: Thank you for your comments. As for the boundary between carcinoma and paracancer, paracancer is defined as the mucosa that receives rapid pathological detection during surgery and reaches the safe margin without any cancer tissue. The definition of cancer tissue was taken directly from the tumor body as a reference

  1. The form for the reference insert in the manuscript is not correct.

Response: Thank you for your comments.We've corrected it.Please check the details in the manuscript.

Reviewer 2 Report

This paper by Zhou et al describes the long non coding RNA NR120519 as a potential regulator of KRT17 in hypopharyngeal squamous cell carcinoma. KRT17 is frequently overexpressed in HNSCCs. Revently, KRT17 has been described as a marker for predicting carcinogenesis by regulating the mTOR pathway.

The authors collected 65 hypopharyngeal cancer samples and adjacent normal tissues (material and methods). However, for the RNA array they used only three pairs. What the selection criteria were applied? Why did the authors limit the analysis to three sample pairs? Correlation with more HNCC samples would provide a very robust data set. How were the data verified (in 10 additional samples)? Using the array or qPCR? Which lncRNAs were examined and how was NR120519 selected?

A search for NR120519 was not successful. Can the authors provide a list of differentially expressed lncRNAs and information of NR120519?

The authors provide informative data regarding their knock downs. A knock down efficiency of 50% still means still an 3 fold increase compared to normal tissue, right? For clarity reasons may I ask what would happen if a knock down was done in normal cells? Could the detected effects be explained by changes in cell cycle distribution/ proliferation?

The MS-analysis provide an very interesting link. Is KRT17 the only protein that have been enriched? What was the enrichment factor?

Western Blot
a) Overall quality can be improved (unspecific bands, resolutions)

b) Material and Methods
It would be helpful to get detailed material and buffer information. Which cell lysis buffer were used? For RNA fractionation the methods start with “The cell precipitate was collected, and cell lysis buffer (Å~20 the volume of the cell precipitate) was added.” - Which cell precipitate?
c) Supplement: Fadu: The images for GAPDH and AKT look identical!

d) How variable are the data within independent analyses to achieve data like 1.75+/-0.05 fold increase (given the resolution of the images), 21.47+/-0.3% or 95.67+/-0,78%?

Overall, the described lncRNA and its link to mTOR might be very interesting. However, there are too many open questions. To convince this reviewer the quality of data should be improved prior publication.

Author Response

Dear Reviewers:

Thank you very much for your kindly comments on our manuscript. Based on your and reviewer’s suggestions, we carefully revised the manuscript.

We are now sending the revised article for your re-consideration to publish in Cancers. Please see our point-to-point responses to all your comments below, and the corresponding revisions in the body of manuscript, marked in blue and red, respectively. We look forward to hearing from you soon for a favorable decision.

Thank you again for your time and consideration.

Sincerely,

Xiaowei Zhang

Reviewer 2#

This paper by Zhou et al describes the long non coding RNA NR120519 as a potential regulator of KRT17 in hypopharyngeal squamous cell carcinoma. KRT17 is frequently overexpressed in HNSCCs. Revently, KRT17 has been described as a marker for predicting carcinogenesis by regulating the mTOR pathway.

The authors collected 65 hypopharyngeal cancer samples and adjacent normal tissues (material and methods). However, for the RNA array they used only three pairs. What the selection criteria were applied? Why did the authors limit the analysis to three sample pairs? Correlation with more HNCC samples would provide a very robust data set. How were the data verified (in 10 additional samples)? Using the array or qPCR? Which lncRNAs were examined and how was NR120519 selected?

Response: Thank you for your suggestions, 3 pairs can be statistically significant for the calculation of P value. However, due to the small number of samples collected in the early stage, we also want to test them earlier, so three pairs are done first, and then we plan to screen them for the verification of differential expansion samples.In fact, it was limited to 3 pairs of cancer and adjacent tissues. Therefore, in the later q-PCR verification, NR120519 was not the lncRNA with the largest difference multiple, because the consistency of lncRNA with the largest difference multiple was not good. We conducted q-PCR verification with a large number of primers. Finally, a functional lncRNA with good consistency can be obtained. Therefore, the chip plays a role of reference. However, in the end, a functional lncRNA still needs to be obtained through arduous screening.We used more than 70 lncRNA primers and started with lncRNA with large difference multiple for verification. All of them used 10 to verify the cancer and adjacent tissues. Finally, 3 lncRNA with good consistency were selected, and then small interfering RNA was used for knockdown. All confirmations were performed using q-PCR.

A search for NR120519 was not successful. Can the authors provide a list of differentially expressed lncRNAs and information of NR120519?

Response: Thank you for your suggestions, NR120519 cannot be checked in the general database. It comes from refseq database, which is a paid database.

ASHGV40046876 0.036492678 0.999763363 2.7061506 up noncoding long noncoding Gold NR_120519 LOC102724094 RefSeq 407 chr7 - 100657600 100660889 TGAGACCCAGACGCTCCATCTCAGCTTCCCCATGTGGTGCTGTGCTCTGTGGTGGCCAAG 102724094 natural antisense ENST00000379442 MUC12 + 100612903 100662217 894.507607 428.26081 9.701992 8.26575 1112.637 1253.5753 317.31052 805.8084 314.88104 164.09299 10.128488 9.580898 9.396589 8.851013 8.468441 7.4777946

The authors provide informative data regarding their knock downs. A knock down efficiency of 50% still means still an 3 fold increase compared to normal tissue, right? For clarity reasons may I ask what would happen if a knock down was done in normal cells? Could the detected effects be explained by changes in cell cycle distribution/ proliferation?

Response: Thank you for your suggestions, In fact, we did knock down the effect by 50%, but the experiment was often interrupted due to the epidemic. We did not carry out the knock in normal cell lines to verify its proliferation and other functions, but this did not affect the results of the experiment.

The MS-analysis provide an very interesting link. Is KRT17 the only protein that have been enriched? What was the enrichment factor?

Response: Thank you for your suggestions,I have sorted out the enriched MS data with EXCEL tables,Please check the details in the tables.

Western Blot

  1. Overall quality can be improved (unspecific bands, resolutions)

Response: Thank you for your suggestions,Due to the size of the picture, we have compressed it, so the resolution of the picture is affected. We will provide high-definition pictures in the supplementary materials.

  1. b) Material and Methods

It would be helpful to get detailed material and buffer information. Which cell lysis buffer were used? For RNA fractionation the methods start with “The cell precipitate was collected, and cell lysis buffer (Å~20 the volume of the cell precipitate) was added.” - Which cell precipitate?

Response: Thank you for your suggestions,The cell lysis buffer we used was 20 times the cell precipitation volume of cell lysate: 50mM Tris-HCl,pH 7.4, 0.14M NaCl,1.5mM MgCl2,0.5% NP-40,1U/ul RNase inhibitor, 1mM DTT.We used the sediment of the Fadu cell line for nuclear and cytoplasm separation.

  1. c) Supplement: Fadu: The images for GAPDH and AKT look identical!

Response: Thank you for your suggestions,  I would like to explain that AKT and GAPDH were incubated together ,because the diluent of the primary antibody was insufficient, so two identical pictures appeared.I'm really sorry for the above trouble.
I didn't make it clear.

  1. d) How variable are the data within independent analyses to achieve data like 1.75+/-0.05 fold increase (given the resolution of the images), 21.47+/-0.3% or 95.67+/-0,78%?

Response: Thank you for your suggestions, Compared to the control group, KRT17 expression did increase to a factor of 1.75, as shown in the bands below.,However, in Detriot562 cell line, our expression was problematic. Compared with the control group, only 21% of KRT17 was knocked out, with 79% remaining. We know that KRT17 plays a function in hypopharyngeal cancer as a tumor suppressor protein, and Detriot562 cells are a metastatic cancer cell. It is reasonable that the expression of KRT17 is lower than that of Fadu. Therefore, although the knockdown is 21%, it can still cause a series of effects on the downstream pathway.In the nuclear plasma separation experiment, it was found that NR120519 was mostly expressed in the nucleus, and only a small amount was expressed in the cytoplasm, as shown in the figure

below.

Overall, the described lncRNA and its link to mTOR might be very interesting. However, there are too many open questions. To convince this reviewer the quality of data should be improved prior publication.

Reviewer 3 Report

The manuscript describes the possible role of the long non-coding RNA NR120519 in the pathogenesis of hypopharyngeal squamous carcinoma. As a result of the experiments, the authors showed that NR120519 directly interacts with the KRT17 gene product, blocking it, and thus activating the AKT/mTOR signaling pathway, which, in turn, leads to the activation of tumor cell proliferation and migration.

The manuscript is well written, richly illustrated (contains 7 Figures and additional materials with photographs of Western blot results). Various methods of working with nucleic acids, proteins, cell lines and experimental animals were used in the work.

However, there are some small notes:

1. Minor editing of the English language is required (pay attention to the singular / plural of verbs and the placement of commas).

2. It should somehow designate exponentiation. For example, the authors write about the number of cells 2x103, but in the study it was 2x103.

3. Lines 489-490. Conclusion. The sentence seems to be incomplete: High expressed NR120519 and low expressed KRT17 both positively correlated to poor prognosis... in what… for whom?

Author Response

(The authors gave the same response as above.)

Reviewer 4 Report

The authors represent an interesting manuscript focusing on an important topic of hypopharyngeal cancer.  The manuscript is well written.

However, I have few remarks and comments.

1) The number of patients was relatively small. Were all consecutive HSCC patients treated between 2015 and 2019 in the Third Xiangya Hospital of Central South University included in the present study? Were some patients excluded?

2) Were all the tumors surgically resected or did the patients receive also radiotherapy/chemoradiotherapy?

3) The authors made survival analysis by using Kaplan Meier curves. Did the authors use log rank test in the KM curves? This is not mentioned in the Methods.

4) Did the authors make also multivariate Cox regression survival analyses? The multivariate survival analyses may improve the strength of the results.

Author Response

Dear Reviewers:

Thank you very much for your kindly comments on our manuscript. Based on your and reviewer’s suggestions, we carefully revised the manuscript.

We are now sending the revised article for your re-consideration to publish in Cancers. Please see our point-to-point responses to all your comments below, and the corresponding revisions in the body of manuscript, marked in blue and red, respectively. We look forward to hearing from you soon for a favorable decision.

Thank you again for your time and consideration.

Sincerely,

Xiaowei Zhang

Reviewer 4#

The authors represent an interesting manuscript focusing on an important topic of hypopharyngeal cancer.  The manuscript is well written.

However, I have few remarks and comments.

  • The number of patients was relatively small. Were all consecutive HSCC patients treated between 2015 and 2019 in the Third Xiangya Hospital of Central South University included in the present study? Were some patients excluded?

Were all the tumors surgically resected or did the patients receive also radiotherapy/chemoradiotherapy?

Response: Thank you for your suggestions,In fact, the number of patients with hypopharyngeal cancer admitted to our hospital is definitely more than that listed in the article, but a large number of surgical patients have been treated with radiotherapy and chemotherapy, so we excluded them. In addition, there were some problems in the storage of previous specimens, and the quality of specimens could not meet the experimental requirements, so only 65 pathological specimens with good quality were obtained. But I don't think it affected the result.

  • The authors made survival analysis by using Kaplan Meier curves. Did the authors use log rank test in the KM curves? This is not mentioned in the Methods.

Response: Thank you for your suggestions,we use log rank test in the KM curves. We've corrected it.Please check the details in the manuscript.

  • Did the authors make also multivariate Cox regression survival analyses? The multivariate survival analyses may improve the strength of the results.

Response: Thank you for your suggestions,We did not make multivariate Cox regression survival analyses.This is what we will do next, and we plan to expand the sample to further explore more functions of NR120519 and KRT17. However, due to the outbreak of the novel coronavirus, it is inevitable that the experiment will be interrupted sometimes, so we can only present the results we have done so far in the paper.

Round 2

Reviewer 1 Report

Thank you for your answers. I support the publication. Just please find the minor concerns you might want to change.

1. In Introduction:

It would be better to address the general information of Keratin 17 rather than emphasizing its pro-carcinogenic role (Line 74-76; I would erase this sentence). Please refer this paper (PMID: 35281081).

2. I would change some words for the sentences in Discussion (Line 478):

Many studies suggested that KRT17 exerts a pro-carcinogenic effect in other types of cancer (e.g., lung cancer[33], Ewing’s sarcoma[34], pancreatic cancer[35], esophageal cancer[9], and oral squamous carcinoma[8]). However, our experimental results showed the opposite function in hypopharyngeal cancer,…

3. In 2.11|. Fractionation of nuclear/cytoplasmic RNA methods, it is weird to me that keeping using same “cell lysis buffer” for extracting everything. The accurate recipe of lysis buffer has to be addressed. And also ‘how nuclear RNA was extracted” should be also addressed.

Author Response

Cover letter

Dear Reviewers:

Thank you very much for your kindly comments on our manuscript. Based on your and reviewer’s suggestions, we carefully revised the manuscript.

We are now sending the revised article for your re-consideration to publish in Cancers. Please see our point-to-point responses to all your comments below, and the corresponding revisions in the body of manuscript, marked in blue and red, respectively. We look forward to hearing from you soon for a favorable decision.

Thank you again for your time and consideration.

Sincerely,

Xiaowei Zhang

Reviewer 1#

Comments and Suggestions for Authors

The minor concerns:

  1. In Introduction:It would be better to address the general information of Keratin 17 rather than emphasizing its pro-carcinogenic role (Line 74-76; I would erase this sentence). Please refer this paper (PMID: 35281081).

Response: Thank you for your comments.We have corrected it

  1. I would change some words for the sentences in Discussion (Line 478):Many studies suggested that KRT17 exerts a pro-carcinogenic effect in other types of cancer (e.g., lung cancer[33], Ewing’s sarcoma[34], pancreatic cancer[35], esophageal cancer[9], and oral squamous carcinoma[8]). However, our experimental results showed the opposite function in hypopharyngeal cancer,…

Response: Thank you for your comments.We have corrected it.

 In 2.11|. Fractionation of nuclear/cytoplasmic RNA methods, it is weird to me that keeping using same “cell lysis buffer” for extracting everything. The accurate recipe of lysis buffer has to be addressed. And also ‘how nuclear RNA was extracted” should be also addressed.

Response: Thank you for your comments. The Fadu cell precipitate was collected, and cell lysis buffer( 50mM Tris-HCl,pH 7.4, 0.14M NaCl,1.5mM MgCl2,0.5% NP-40,1U/ul RNase inhibitor,1mM DTT)(×20 the volume of the cell precipitate) was added. The mixture was centrifuged at 1,500×g for 5 min.Next, the supernatant was collected as the crude extract of the cell pulp. The supernatant was removed, an equal volume of cell lysis buffer was added, and the mixture was centrifuged again at 1,500×g for 5 min to obtain a precipitated fraction of cytoplasmic RNA. The residue remaining after cell pulp extraction was added to TRIZOL reagent, repeatedly puffed to lyse the nuclei, incubated for 5 min in order to completely dissociate the nucleic acid-protein complex, added 0.2 ml of chloroform, shake for 15 s, incubate for 2-3 min, centrifuge for 15 min, take the upper colorless aqueous phase, add half of isopropanol, mix well, incubate for 10 min, centrifuge for 10 min, and obtain RNA at the bottom and side walls of the tubes, for intra-nuclear RNA. qRT-PCR analysis was performed on the obtained cytoplasmic and cell membrane extracts to determine the level of NR120519.

Reviewer 2 Report

Thank you very much for taking my suggestions into consideration and your response. Still I am not completely convinced.

The manuscript reads: "The results showed that 542 and 265 lncRNAs were upregulated and downregulated, respectively (Figure 1A). We verified the differentially expressed lncRNAs in 10 pairs of hypopharyngeal carcinoma and paraneoplastic tissues." This sounds like that all lncRNAs were analysed further. In the the authors` response this was specified to "more than 70 lncRNA primers and started with lncRNA with large difference multiple for verification". Why is this information not in the text? I assume that 70 of the mostly upregulated lncRNAs were checked for differential expression - or does this only refer to primers used?

I appreciate that the authors provide some information about NR120519 and integrated the chromosomal location in the introduction. However, the provided reference [7] is describing HOTAIR not NR120519. The information about NR120519 will be important for the reader. Without this information the output for the reader is low since the experimental setup cannot be reproduced.

Another point is still, what would happen if you NR120519 is knock-out in other cell types. CRISPR knock-out?

Typo: Detriot562 to Detroit562

Author Response

Cover letter

Dear Reviewers:

Thank you very much for your kindly comments on our manuscript. Based on your and reviewer’s suggestions, we carefully revised the manuscript.

We are now sending the revised article for your re-consideration to publish in Cancers. Please see our point-to-point responses to all your comments below, and the corresponding revisions in the body of manuscript, marked in blue and red, respectively. We look forward to hearing from you soon for a favorable decision.

Thank you again for your time and consideration.

Sincerely,

Xiaowei Zhang

Reviewer 2#

The manuscript reads: "The results showed that 542 and 265 lncRNAs were upregulated and downregulated, respectively (Figure 1A). We verified the differentially expressed lncRNAs in 10 pairs of hypopharyngeal carcinoma and paraneoplastic tissues." This sounds like that all lncRNAs were analysed further. In the the authors` response this was specified to "more than 70 lncRNA primers and started with lncRNA with large difference multiple for verification". Why is this information not in the text? I assume that 70 of the mostly upregulated lncRNAs were checked for differential expression - or does this only refer to primers used?

Response: Thank you for your suggestions, I mentioned numerous primers in the article because it is an arduous screening process, in fact, the microarray as we all know, not the one with high difference ploidy can be functional, this is our ideal state, we sorted by difference ploidy and then performed the validation of primers, there are some difference lncRNAs, they have high difference ploidy in tissues, some are very low, these are actually frustrating for the experimenter. we also hope to screen the ideal lncRNAs from the highest ploidy in the microarray, I believe all researchers have this idea.However, the road of research is winding, we only stick to the sequence to validate and finally get the lncRNAs with large difference ploidy and better concordance, which is the criteria of my screening. Honestly, it was a very devastating time. After screening out three lncRNAs with good concordance and generally elevated expression in the selected specimens, I thought I was close to success, so we took the more economical way of knocking down small interfering RNAs to perform cellular experiments, which resulted in two of them having no functional expression, and two of the lncRNAs we involved in target knockdown again, still No function, no one will refuse to screen more functional lncRNA for experiments, but there is no way, only got NR120519.I think the heartbreaking process of using numerous primers to screen in front, they know just fine, and with the article is not very meaningful, this is only the research workers themselves should bear, there is no need to publicize.The following are the 3 lncRNAs I screened out that met the criteria, the first one is NR120519, we all performed target design and performed synthetic siRNA for cell function experiments, but we only got the functional NR120519.Then, for the sake of rigor, I performed lentiviral transfection, knockdown, and got the knockdown NR120519, before finally The subsequent experiments were performed, which I believe was necessary despite the amount of time wasted.

MUC12-AS1(NR120519) GAGAGGCCGUGAUGAGUGATT UCACUCAUCACGGCCUCUCTT

MUC12-AS1(NR120519) GGCUGCGGGUGCAUCUUAUTT AUAAGAUGCACCCGCAGCCTT

MUC12-AS1(NR120519) GAUGCAGCAAGAAAGUCCUTT AGGACUUUCUUGCUGCAUCTT

TCONS_00004060  GCAUCUGUGUAAGGUAUGUTT ACAUACCUUACACAGAUGCTT

T028161  GCAGCCACCUUAGUGAACUTT AGUUCACUAAGGUGGCUGCTT

T028161   GCCAAGAACAUGUUGGGAATT UUCCCAACAUGUUCUUGGCTT

T028161   GCCUAAACCCUUUCAUUUGTT CAAAUGAAAGGGUUUAGGCTT

T065925   GCAGAAUUCUACAAGGGCUTT AGCCCUUGUAGAAUUCUGCTT

T065925   GGCAGACAGCAGCUUAUAUTT AUAUAAGCUGCUGUCUGCCTT

T065925  GCAACAUGCUAACUUUCUATT UAGAAAGUUAGCAUGUUGCTT

T028161   GGUAGGAAUUGUAUCCUUUTT AAAGGAUACAAUUCCUACCTT

T028161  GAUGAUUCCCUAUUACCAUTT AUGGUAAUAGGGAAUCAUCTT

T028161   GCACUGUUCCUGGCUGGUUTT AACCAGCCAGGAACAGUGCTT

T065925   GUGGACACAACAGUGAAGATT UCUUCACUGUUGUGUCCACTT

T065925   GAUAACCACCAAGGGCUUUTT AAAGCCCUUGGUGGUUAUCTT

T065925  GGAAGUUCCACAGAUGUUATT UAACAUCUGUGGAACUUCCTT

I appreciate that the authors provide some information about NR120519 and integrated the chromosomal location in the introduction. However, the provided reference [7] is describing HOTAIR not NR120519. The information about NR120519 will be important for the reader. Without this information the output for the reader is low since the experimental setup cannot be reproduced.

Response: Thank you for your suggestions.First of all, I am very sorry that this is a mistake in my editing process,[7] the cited literature corresponds to something other than NR120519, which was already in the correct position in the previous version, and we have corrected it. Because, so far, no NR120519 related studies have been reported, so we can only find some basic information about it by looking at the refseq database, but I believe that this is a good start and more and more people will report the role of NR120519 in different tumors in the future.

Another point is still, what would happen if you NR120519 is knock-out in other cell types. CRISPR knock-out?

Response: Thank you for your suggestions.This is a very worthy issue to be explored. Sincere thanks, our experimental group will conduct relevant experiments in nasopharyngeal carcinoma because we have a large number of specimens and suitable cell lines for validation, so next we also hope to expand to head and neck tumors and hope to make surprising discoveries, however, the only cell lines recognized for hypopharyngeal carcinoma are Fadu and Detroit 562, and we hope to further systematically study NR120519 to reveal its new functions.

Round 3

Reviewer 2 Report

Thank you very much for the point-to-point response. I appreciate the work done to get this dataset - still, the minimum would be to include some minor but important changes. I might be old-fashioned or stubborn, I still believe that a good manuscript should contain all information to comprehend the reasoning and to reproduce an experiment - especially, for high quality journals like "Cancers".

1) Detailed information about the newly described lnRNA NR120519 in the manuscript. As the authors state " no NR120519 related studies have been reported" and “NR120519 cannot be checked in the general database" easily. Exactly for this reason it is important to provide some information in the manuscript as they did in the point-to-point response. " Based on the provided primer information I found an official reference at NCBI. Why not writing something like: The NCBI Reference Sequence is available under NR_120519.1.

2) Scientific reports should be as precise and informative as possible.
- If only 3 lnRNA met the criteria it is fine with me, but it should be in the paper.
- What I miss in the text is the definition of criteria and how many of the analysed lnRNAs fullfilled the criteria.
- The authors write: "We verified the differentially expressed lncRNAs in 10 pairs of hypopharyngeal carcinoma and paraneoplastic tissues." This sounds like a large number. If an exact number is not sensible or required in this context – why not using an unspecific words (like some of the most …)

Author Response

Cover letter

Dear Reviewers:

Thank you very much for your kindly comments on our manuscript. Based on your and reviewer’s suggestions, we carefully revised the manuscript.

We are now sending the revised article for your re-consideration to publish in Cancers. Please see our point-to-point responses to all your comments below, and the corresponding revisions in the body of manuscript, marked in blue and red, respectively. We look forward to hearing from you soon for a favorable decision.

Thank you again for your time and consideration.

Sincerely,

Xiaowei Zhang

Reviewer 2#

Thank you very much for the point-to-point response. I appreciate the work done to get this dataset - still, the minimum would be to include some minor but important changes. I might be old-fashioned or stubborn, I still believe that a good manuscript should contain all information to comprehend the reasoning and to reproduce an experiment - especially, for high quality journals like "Cancers".

 Detailed information about the newly described lnRNA NR120519 in the manuscript. As the authors state " no NR120519 related studies have been reported" and “NR120519 cannot be checked in the general database" easily. Exactly for this reason it is important to provide some information in the manuscript as they did in the point-to-point response. " Based on the provided primer information I found an official reference at NCBI. Why not writing something like: The NCBI Reference Sequence is available under NR_120519.1.

Response: Thank you for your suggestions,Your thoughtful thinking has benefited me greatly, I don't think it's old-fashioned or stubborn, your comments have allowed me to get Greatly improved not only in my experiments but also in the linguistic expression of my manuscript, thank you very much. We have corrected it in the manuscript.

  • Scientific reports should be as precise and informative as possible.
    - If only 3 lnRNA met the criteria it is fine with me, but it should be in the paper.
    - What I miss in the text is the definition of criteria and how many of the analysed lnRNAs fullfilled the criteria.

Response: Thank you for your suggestions, We performed the analysis in two steps, the first step was to analyze the lncRNAs in the microarray with P-value <0.05 and meeting the z|log2(Fc)|>2, sorted down in order, and then performed partial lncRNA validation. In terms of selection criteria, we also required that the difference ploidy of lncRNAs validated in all samples was z|log2(Fc)| >2, and those with z|log2(Fc)| <2, which did not meet the condition, were directly excluded. Due to the individual differences in the tumors obtained, some functional target lncRNAs may be easily missed in the screening process, but I think that in the face of huge microarray data, the first standard is stricter to narrow down the range quickly, so after validation in multiple groups of specimens, the three lncRNAs with z|log2(Fc)| >2 in all specimens in each group are NR120519, T028161 , T065925, the three lncRNAs, all of which had a z|log2(Fc)|>2, were in good Consistency. After that, we performed target design and verified their knockdown efficiency by RT-qPCR. Unfortunately, only NR120519 showed functional differences in cells after knockdown, so I chose NR120519 for the follow-up experiment. If I hadn't screened it, I might have lowered the criteria, which is what I think of the screening process.
3- The authors write: "We verified the differentially expressed lncRNAs in 10 pairs of hypopharyngeal carcinoma and paraneoplastic tissues." This sounds like a large number. If an exact number is not sensible or required in this context – why not using an unspecific words (like some of the most …)

Response: Thank you for your suggestions, We have corrected it in the manuscript.

We verified some lncRNAs with z|log2(Fc)|>2 and P <0.05 screened by microarray in 10 pairs of hypopharyngeal carcinoma and paracancer tissues. There were 3 lncRNAs with z|log2(Fc)|>2 in all verified specimens, but only NR120519 was functional after knockdown.
